# The Intake of Ultra-Processed Foods and Prevalence of Chronic Kidney Disease: The Health Examinees Study

**DOI:** 10.3390/nu14173548

**Published:** 2022-08-28

**Authors:** Anthony Kityo, Sang-Ah Lee

**Affiliations:** 1Department of Preventive Medicine, College of Medicine, Kangwon National University, 1 Gangwondeahakgil, Chuncheon-si 24341, Korea; 2Interdisciplinary Graduate Program in Medical Bigdata Convergence, Kangwon National University, 1 Gangwondeahakgil, Chuncheon 24341, Gangwon, Korea

**Keywords:** ultra-processed food, glomerular filtration rate, kidney function, FFQ

## Abstract

Emerging evidence links several health outcomes to the consumption of ultra-processed food (UPF), but few studies have investigated the association between UPF intake and kidney function. This cross-sectional study investigated the prevalence of chronic kidney disease (CKD) in relation to UPF intake in Korea. Data were obtained from the 2004–2013 Health Examinees (HEXA) study. The intake of UPF was assessed using a 106-item food frequency questionnaire and evaluated using the NOVA classification. The prevalence of CKD was defined as an estimated glomerular filtration rate (eGFR) of <60 mL/min/m^2^. Poisson regression models were used to compute the prevalence ratios (PR) of CKD according to quartiles of the proportion of UPF intake (% food weight). A total of 134,544 (66.4% women) with a mean age of 52.0 years and an eGFR of 92.7 mL/min/m^2^ were analysed. The median proportion of UPF in the diet was 5.6%. After adjusting for potential confounders, the highest quartile of UPF intake was associated with the highest prevalence of CKD (PR 1.16, 95% CI 1.07–1.25), and every IQR (6.6%) increase in the proportion of UPF in the diet was associated with a 6% higher prevalence of CKD (PR 1.06, 95% CI 1.03–1.09). Furthermore, the highest consumption of UPF was inversely associated with eGFR (Q4 vs. Q1: β −1.07, 95% CI −1.35, −0.79; per IQR increment: (β −0.45, 95% CI −0.58, −0.32). The intake of UPF was associated with a high prevalence of CKD and a reduced eGFR. Longitudinal studies in the Korean population are needed to corroborate existing findings in other populations.

## 1. Introduction

Globally, the number of chronic kidney disease (CKD)-related deaths has increased over the past 20 years, and the number of people affected by CKD is projected to increase, which calls for urgent public health preventive measures to combat this burden [1]. The rise in the global burden of CKD is mainly attributed to diabetes, hypertension, high body mass index, ageing, infections, and environmental toxins [2]. However, modifiable lifestyle factors, such as diet, are continuously being recognised as etiological and prognostic factors of CKD [3]. Recent systematic reviews and meta-analyses of observational studies have reported that a healthy dietary pattern is associated with a lower risk of CKD [4,5,6], whereas a Western dietary pattern and unhealthy dietary patterns are associated with a higher risk of CKD [4,6]. Furthermore, an unhealthy diet, as assessed by the Alternate Healthy Eating Index 2010 (AHEI-2010), was associated with kidney function decline among Hispanics/Latinos [7].

The Western diet and unhealthy diets are characterised by components of a recently defined pattern: the ultra-processed food (UPF) pattern. UPFs, defined by the NOVA classification as foods formulated mostly or entirely from food-derived substances containing little or none of the original food form [8,9], are becoming increasingly popular in diets globally [10,11]. The energy contribution of UPF to the total diet has been documented from 25.1% in Korea [12,13] and 45% in Canada [14] to 57.9% in the US [15]. 

UPFs are high in energy, sugars, and trans-fats and are deficient in dietary fiber, various micronutrients, and other bioactive compounds [9]. They contain fractions of whole foods or chemically modified substances, such as high-fructose corn syrup, hydrogenated oils, and hydrolysed proteins, and food additives, such as thickeners and emulsifiers, that are rarely or never used in the kitchen [8]. Thus, UPF intake has been linked to several non-communicable diseases and mortality [16]. A few longitudinal studies have reported a positive association between UPF intake and CKD in the Netherlands [17], Spain [18], and the US [19]. These studies have been conducted in Western countries with high intakes of UPF. The association between UPF intake and health outcomes in Asian populations with low intakes of UPF and diverse dietary habits is largely unknown. This study assessed the prevalence of CKD in relation to ultra-processed food intake. The secondary aim was to assess the relationship between UPF intake and estimated glomerular filtration rate (eGFR) using baseline data from the Health Examinees (HEXA) study.

## 2. Materials and Methods

### 2.1. Study Population

The HEXA study is a prospective cohort within the Korean Genome and Epidemiology study (KoGEs) [20]. The study was conducted to investigate the epidemiologic characteristics, genomic features, and gene–environment interactions of major chronic diseases in the Korean population. A total of 173,357 participants aged 40 years and older were recruited between 2004 and 2013 at 38 health examination centres and training hospitals located in the 8 regions of Korea. Details of the HEXA study have been published elsewhere [21]. In the present analysis, we excluded participants from invalid recruitment sites that had limited equipment for quality control of biological specimens (*n* = 31,375), those who withdrew from the study (*n* = 11), and those aged above 69 years (*n* = 2626) to obtain the HEXA-Gem sample, which comprised 139,345 participants. We applied additional exclusion criteria and retained an analytical sample of 134,544 (Figure 1). The Institutional Review Board of Seoul National University Hospital approved this study for statistical analysis (IRB No. E-1503-103-657).

### 2.2. Assessment of UPF Intake

The intake of UPF was assessed using a 106-item semiquantitative food frequency questionnaire (FFQ) developed and validated for the KoGES [22] and classified on the basis of the degree and extent of processing using the NOVA classification [9]. We employed a slightly modified four-stage approach developed by Khandpul and colleagues to assign each food item to a NOVA class [23]. In brief, AK independently assigned all the 106 items to the NOVA groups and then consulted S-AL, who participated in the design of the HEXA study, to validate AK’s classification. Regarding foods for which a consensus could not be reached, we visited food stores and checked the website to verify the food labelling information and manufacturing processes, respectively. We additionally referred to previous publications and checked their UPF categorisation [9,13]. For mixed dishes and aggregated food groups which contain food items with different degrees of processing, we disaggregated them and applied weights using the food recipe information. The applied weights represented the % weight contributed by a food item to the food group or dish. Finally, the UPF items in this study comprised instant noodles, breakfast cereals, breads, bread spreads (jam, butter, and margarine), cakes, cookies, crackers, snacks, candies and chocolate, pizza and hamburgers, processed red meat (ham and sausage), processed fish, flavoured milk, yoghurts, ice cream, soymilk drink, soft drinks and fruit sodas, sweet rice punch, and tomato ketchup (Appendix A). We computed the percentage of the total food contributed by UPF (% food weight from UPF) and categorised participants into quartiles of % food weight from UPFs. 

### 2.3. Assessment of CKD [24]

Serum creatinine was used to compute the estimated glomerular filtration rate (eGFR) following the Chronic Kidney Disease Epidemiology Collaboration equation (CKD-EPI). The prevalence of CKD was defined as eGFR < 60 mL/min/1.73 m^2^.

### 2.4. Assessment of Covariates

Baseline information about the demographic, lifestyle, physical, and clinical characteristics of participants was collected using an interviewer-administered questionnaire. The following socio-demographic covariates were evaluated: marital status (married or single); education level (≤ elementary school, middle school, high school, or ≥ university); and monthly family income (<1000, 1000–3000, or ≥3000 USD). Regarding lifestyle variables, current smokers were defined as participants who had smoked more than 400 cigarettes during their lifetime and were still smoking [25]. Drinking was evaluated as current or non-current drinkers. Non-current drinkers included past drinkers and those who had never drunk alcohol. Participants were asked to report (1) whether they engaged in regular physical exercise that caused body sweating; (2) the number of times they engaged in these exercises in a week (1–2 times/week to every day); and (3) the duration of the exercise. Regular exercise was defined as engaging in activities that caused body sweating at least 5 times a week lasting at least 30 minutes. Body mass index (BMI) was calculated as weight in kilograms divided by the square of height in meters (kg/m^2^). Diabetes was defined as a fasting blood glucose level ≥126 mg/dL or self-reported drug treatment for elevated fasting blood glucose. Hypertension was defined as a systolic blood pressure ≥130 mmHg, a diastolic blood pressure ≥85 mmHg, or self-reported drug treatment for elevated blood pressure. Cardiovascular disease (CVD) was defined as a self-reported diagnosis of myocardial infarction or ischemic stroke or the intake of medication to treat these conditions.

### 2.5. Statistical Analysis

SAS software version 9.4 (SAS Institute Inc., Cary, NC, USA) was used to analyse the data, and *p* < 0.05 was used to define statistical significance. The distribution of participant characteristics according to quartiles of UPF intake was described using percentages for categorical variables and least-square means for continuous variables. We created a category of missing data on income (14.7%); missing data on education (1.85%), marital status (1.03%), smoking (0.95%), drinking (0.89%), physical exercise (0.81%), high blood pressure (0.14%), and high blood glucose (0.08%) were replaced by the mode; and missing data on BMI were replaced by the median value of BMI (23.7). 

To determine the prevalence of CKD according to UPF intake, we estimated prevalence ratios (PR) and their corresponding 95% confidence intervals (CI) using modified Poisson regression with the generalised estimating equation implemented in the Genmod procedure. The first model was adjusted for age (continuous), sex, and total energy intake (continuous). The second model was further adjusted for education, income level, smoking, drinking, physical exercise, and BMI (continuous), and the third model was further adjusted for high blood pressure, high blood sugar, and CVD. In the secondary analysis, we used quantile regression to compute regression coefficients and their 95% CIs for the association between quartiles of UPF intake and eGFR using Q1 as the reference group. Quantile regression estimates the conditional median (or other percentile) instead of the mean of the dependent variable and has the advantage of being robust regarding the distribution of the dependent variable.

We calculated the interquartile range (IQR) of UPF intake and divided the original UPF variable by the IQR (% food weight from UPF/IQR of % food weight from UPF). We then included the resulting variable in the model as a continuous variable to estimate the PR of CKD and the regression estimates for eGFR per IQR increment of UPF intake. For exploratory analyses, we stratified the analysis by age, sex, smoking, drinking, physical exercise, BMI, high blood glucose, high blood pressure, and prevalent CVD. We tested whether these variables modified the association between UPF and kidney function (eGFR) by including cross-product terms between UPF (IQR) and each stratum variable and checking the *p*-value (*p* for interaction) of the cross-product term. We also estimated the regression coefficients and their 95% confidence intervals for the relationship between UPF sub-groups (per g/day) and eGFR using quantile regression adjusted for variables in model 3 of the main analysis. The regression coefficients represent the change in the median eGFR per 1 g increment of the intake of each UPF sub-group.

Finally, we conducted sensitivity analyses by (1) excluding participants with high blood glucose, high blood pressure, or hypertension; (2) adjusting for dietary fat, phosphorous and cholesterol intake; (3) restricting the analysis to participants with complete data on confounding variables; and (4) including coffee cream, cheese, and ready-to-eat grain mixes/porridges in the calculation of UPF intake.

## 3. Results

More than 66% of the participants included in this study were women. The mean age, BMI, and eGFR of participants were 52.0 years, 23.8 kg/m^2^, and 92.0 mL/min/m^2^, respectively. The median per cent of weight contributed by UPF was 5.6%. Young age, female sex, high education, and income level were prevalent among participants with the highest intake of UPF (Table 1). 

In addition, the highest consumers of UPF were more likely to smoke and drink, but less likely to engage in regular exercise. The highest consumers of UPF were less likely to be overweight and to have high blood pressure, high blood glucose, and CVD. A high intake of total energy, fat, and dietary cholesterol but a lower intake of carbohydrates, fibre, and sodium were observed among highest consumers of UPF.

The PRs and 95% confidence intervals of CKD according to quartiles of % food weight from UPF are displayed in Table 2. After adjusting for potential confounders (model 3), increased intake of UPF was characterised by a corresponding increase in the prevalence of decreased kidney function (*p* for trend = 0.003). In addition, every IQR increase in the % weight of UPF was associated with a 6% increase in the prevalence of decreased kidney function (PR 1.06, 95% CI 1.03–1.09).

Table 3 shows the association between UPF intake and eGFR. Participants with the highest intake of UPF (Q4) had lower eGFRs than those with the lowest intake of UPF (β −1.04, 95% CI −1.31, −0.77). Every IQR increment in UPF intake was associated with a reduction in eGFR (β −0.45, 95% CI −0.58, −0.32).

The stratified analyses of the association between UPF intake and eGFR are shown in Table 4. The inverse relationship between UPF intake (per IQR increment) and eGFR was stronger in men (β = −0.51, *p* interaction = <0.001), current smokers (β = −0.81, *p* interaction < 0.001), the obese (β = −0.39, *p* interaction = 0.002), those who did not engage in regular exercise (β = −0.47, *p* interaction = 0.002), and those with high blood glucose (β = −0.62, *p* interaction = 0.025). 

Every 1 g/day increase in the intake of ultra-processed breads, cereals and snacks, candies and chocolate, bread spreads, pizza and burgers, milk, yoghurt, ice cream, soybean milk drink, and soft drinks and fruit sodas was associated with a lower eGFR (Table 5). 

The highest consumption of UPF was associated with a high prevalence of CKD even after excluding participants with high blood glucose, high blood pressure, and CVD (PR 1.19, 95% CI 1.05–1.36); adjusting for total fat, cholesterol, and phosphorus intake (PR 1.11, 95% CI 1.03–1.21); excluding participants with missing data on confounding variables (PR 1.12, 95% CI 1.03–1.22); and including ultra-processed cheese, pre-cooked grain mixes (porridges), and coffee cream in the calculation of UPF (PR 1.17, 95% CI 1.08–1.26) (Appendix A).

## 4. Discussion

We analysed data from a large population-based study to investigate the association between the prevalence of CKD, eGFR, and consumption of UPF. We found a high prevalence of CKD among the highest consumers of UPF and an inverse association between UPF intake and eGFR. Furthermore, we found evidence suggesting that sex, smoking, physical exercise, BMI, and high blood glucose modified the association between UPF intake and kidney function.

The contribution of UPF to the diet in the Korean population has been reported at 25.1% in terms of total energy [12,13], which is low compared with the proportions of 45% in Canada [14], 56.8% in the UK [26], and 57.9% in the US [15]. This suggests that Korean dietary patterns, although at a slow pace, are tending towards Western dietary patterns. It is projected that the sales of UPF in Southeast and East Asian countries will approach those of high-income countries by 2035 [9]. The median % weight of UPF in the current study was 5.6%, which is lower than that reported by a previous study in Korea [12]. The previous study estimated UPF intake from 24-hour dietary recall data, expressed UPF intake as a percentage of total energy, included distilled alcoholic beverages in the calculation of UPF, and used dietary data from 2016 to 2018. 

Despite the low intake of UPF reported in the current study, our findings are consistent with previous studies in other populations. Garcia et al. reported a 74% increased odds of renal function decline in older adults among the highest consumers of UPF [18]. A recent cohort study using data from the Atherosclerosis Risk in Communities (ARIC) study reported that participants in the highest quartile of UPF intake were 24% more likely to develop CKD after 24 years of follow-up, and substituting minimally processed foods for UPF was associated with a reduced risk of CKD [19]. The intake of UPF was associated with a decline in eGFR in the Netherlands [17]. UPF has also been previously associated with CKD risk factors [16,27,28]. 

Unlike the cited studies, we found that dietary sodium, total carbohydrates, and fibre intake were low among the highest consumers of UPF in this population. However, the highest consumers of UPF reported a low consumption of vegetables and *Kimchi* (fermented vegetables) and a high intake of dietary cholesterol and total fat. This suggests unique dietary profiles of UPF consumers in the Korean population and that some nutrients, such as sodium and sugars, are not the only mechanisms by which UPFs exert detrimental effects on renal function. The intake of fibre, sodium [5,29,30,31], and sugar have been associated with renal function. In a recent study using a Mendelian randomisation approach, relative fat intake was causally associated with incident CKD [32], but higher vegetable intake was associated with higher eGFR [33]. 

Nutrient and non-nutrient additives in processed foods have been suggested as contributors to kidney function decline. High sodium intake directly alters the renal and vascular system functions independent of blood pressure by increasing oxidative stress [34]. Sodium also exerts a direct effect on the endothelium via changes in shear stress, which modulates the production of transforming growth factor β1 (TGFβ1) and nitric oxide, resulting in vascular and glomerular fibrosis [35]. Sodium additives constitute two-thirds of the total daily salt in Western diets [36] and are present in processed breads, cereals, grains, meats, sauces, and canned items [37]. However, low sodium intake was observed in the highest consumers of UPF in the current study, which is not surprising, given that the major sources of sodium in the Korean population are Kimchi (fermented vegetables), fermented soybean paste, and cooked dishes [38,39]. It should be acknowledged that sodium intake cannot be accurately estimated using dietary recall methods in Korea, unlike in Western countries. This is because the primary sources of sodium intake for Koreans are cooked dishes, and the amount of ingredients and seasonings added is difficult to identify in the cooked foods [38]. Thus, there is a need to conduct studies to validate the FFQ used in this study against a gold-standard method, such as 24 h urinary sodium excretion. 

Inorganic phosphate additives are also largely utilised as pH stabilisers, emulsifiers, and leavening and anti-bacterial agents in breads, meats, and sodas [40,41]. Direct toxicity of increased phosphate concentration is suggested in CKD. Furthermore, high phosphorous intake causes vascular and nephron calcification and renal tubular necrosis in animal models, and renal calcification and albuminuria in incident CKD and end-stage renal disease (ESRD) have also been reported in human studies [42,43].

UPFs contain artificial sweeteners; additives, such as thickeners, emulsifiers, and flavour enhancers; and colourants [8]. Noncaloric artificial sweeteners (NASs) in UPFs, such as saccharin, have been shown to induce glucose intolerance and have been associated with central obesity and poor glycaemic control [44]. Neo-formed substances in processed foods, such as trans-fatty acids, furans, and contaminants [45], and increased exposure to endocrine-disrupting chemicals found in packaging materials [46] may predispose consumers to inflammation and chronic diseases [47].

Although we used a large sample size, adjusted for potential confounders, and used a validated FFQ to assess dietary intake, several limitations should be considered. First, we used a single measure of creatinine to diagnose CKD. The prevalence of CKD varied depending on whether a single eGFR value or repeated eGFR values were used in a population-based cohort [48]. However, the different diagnostic algorithms produced cohorts with the same prognosis. Moreover, cystatin c was shown to be a better predictor of eGFR than creatinine [49]. Unfortunately, cystatin c was not available in the HEXA data at the time of analysis. Nevertheless, we adjusted and stratified for age and BMI to control for the influence of muscle mass and age on serum creatinine. In addition, eGFR estimated from creatinine was shown to accurately predict CKD-associated risk factors [49]. Second, the FFQ used in this study was not designed to assess UPF intake; the identification of specific food items from dishes or aggregated food items was limited, and detailed descriptions of some food items were not available in the FFQ. Therefore, it is possible that some food items were misclassified. Nevertheless, the estimates remained stable when more food items were included in the UPF category. In addition, the dietary data were self-reported, which increases the possibility of underreporting the true intake of UPF. Nevertheless, we excluded implausible dietary reporters to minimise bias due to dietary misreporting. Third, demographic, lifestyle, and disease history were self-reported. Although we adjusted for these factors in the model, residual confounding and misclassification bias could have affected the results. Lastly, this was a cross-sectional study, and temporal relationships can only be inferred. 

## 5. Conclusions

In conclusion, our study found a high prevalence of CKD and a low eGFR among the highest consumers of UPF. Longitudinal studies in this population are needed to corroborate the evidence from existing longitudinal cohort studies. 

## Figures and Tables

**Figure 1 nutrients-14-03548-f001:**
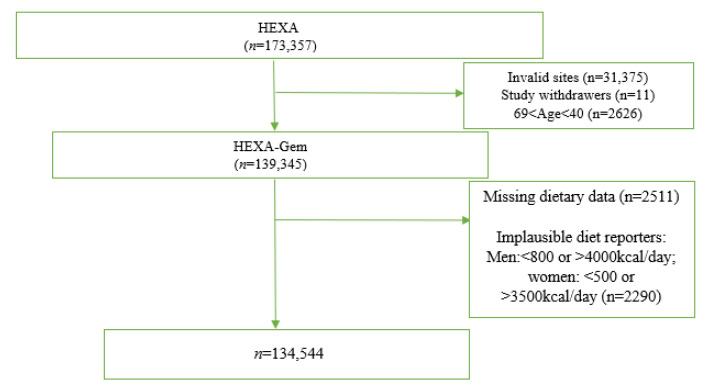
Flow chart showing how study participants were selected.

**Table 1 nutrients-14-03548-t001:** Baseline characteristics of study participants by quartiles of UPF intake.

		Quartiles of UPF Intake, % of food Weight			
Characteristics		Q1	Q2	Q3	Q4
	*n*	33,635	33,637	33,637	33,635
Age		55.1 ± 0.04	52.4 ± 0.04	51.7 ± 0.04	51.7 ± 0.04
Sex	Male	33.6	33.6	33.6	33.6
Education	Elementary	24.0	14.9	12.0	10.5
	Middle	19.6	16.4	14.6	13.5
	High	37.2	41.4	41.3	41.2
	≥College	19.3	27.3	32.1	34.9
Married		88.7	90.4	90.3	88.3
Income, USD	Unknown	15.2	13.0	13.2	13.8
	<1000	13.7	8.8	7.2	7.1
	1000–3000	40.0	38.2	36.4	35.3
	≥3000	31.2	40.0	43.2	43.8
Current smoker		11.8	12.4	12.5	12.2
Current drinker		41.3	46.5	45.8	44.7
Regular physical exercise		49.6	47.4	45.8	43.8
BMI, kg/m^2^		24.0 ± 0.02	24.0 ± 0.02	23.9 ± 0.02	23.6 ± 0.02
BMI categories	<18.5	1.6	1.7	1.8	2.1
	18.5~23.0	35.1	37.6	39	41.4
	23.0~25.0	28.9	27.9	27.6	27.0
	≥25.0	34.4	32.9	31.6	29.5
CKD		4.7	3.8	3.9	4.0
CVD		2.9	2.3	1.8	1.9
High blood glucose		10.4	7.5	6.0	5.4
High blood pressure		47.8	42.3	39.5	38.4
Total energy intake, kcal/day		1571.5 ± 2.61	1684.1 ± 2.58	1806.3 ± 2.58	1884.3 ± 2.58
UPF, % of total food, median (IQR)		1.7 (1.1–2.3)	4.1 (3.5–4.8)	7.4 (6.5–8.4)	13.0 (11.0–16.0)
Unprocessed foods, % of total food		95.1 (93.2–96.6)	91.4 (89.3–93.1)	87.5 (85.0–89.4)	80.6 (76.2–84.0)
Carbohydrates, % of energy		74.0 ± 0.04	72.3 ± 0.04	70.8 ± 0.04	69.6 ± 0.04
Protein, % energy		13.1 ± 0.01	13.4 ± 0.01	13.7 ± 0.01	13.6 ± 0.01
Fat, % energy		11.4 ± 0.03	13.2 ± 0.03	14.6 ± 0.03	16.1 ± 0.03
Fibre, g/day		6.1 ± 0.01	5.8 ± 0.01	5.7 ± 0.01	5.4 ± 0.01
Dietary cholesterol, mg/day		150.2 ± 0.51	161.6 ± 0.5	171.5 ± 0.5	177.9 ± 0.51
Dietary sodium, mg/day		2641.7 ± 6.58	2511.5 ± 6.43	2487.8 ± 6.45	2338.6 ± 6.51
Dietary phosphorus, mg/day		10.0 ± 0.02	9.9 ± 0.02	10.0 ± 0.02	9.8 ± 0.02

Values are means ± SE or %. BMI: body mass index, eGFR: estimated glomerular filtration rate, CVD: cardiovascular disease.

**Table 2 nutrients-14-03548-t002:** Prevalence ratios of decreased kidney function (eGFR < 60) according to UPF intake.

	Quartiles of UPF Intake, % Food Weight					UPF, Continuous
	Q1	Q2	Q3	Q4	*p* for Trend	Per IQR Increment
Cases, *n*	1566	1272	1341	1359		
Model 1	1.00	1.00 (0.93–1.08)	1.15 (1.07–1.24)	1.16 (1.08–1.25)	<0.001	1.06 (1.03–1.09)
Model 2	1.00	1.00 (0.92–1.07)	1.13 (1.04–1.21)	1.13 (1.05–1.22)	0.008	1.05 (1.02–1.08)
Model 3	1.00	1.01 (0.93–1.08)	1.15 (1.06–1.24)	1.16 (1.07–1.25)	0.003	1.06 (1.03–1.09)

Values are prevalence ratios (PR) and 95% CIs unless otherwise specified. Model 1: Adjusted for age, sex, and total energy intake. Model 2: model 1 + educational level, income level, smoking, drinking, physical exercise, and BMI. Model 3: model 2 + high blood pressure, high blood sugar, and prevalent CVD. IQR: interquartile range = 6.6.

**Table 3 nutrients-14-03548-t003:** Regression estimates and 95% CI of eGFR by quartiles and per IQR of UPF intake.

	Quartiles of UPF Intake, % of Food Weight				UPF, Continuous
	Q1	Q2	Q3	Q4	Per IQR Increment
Cases, *n*	1566	1272	1341	1359	
Model 1	Ref	−0.03 (−0.04, −0.02)	−0.04 (−0.05, −0.03)	−0.04 (−0.05, −0.03)	−0.02 (−0.03, −0.01)
Mode 2	Ref	−0.36 (−0.56, −0.16)	−0.69 (−0.91, −0.46)	−1.04 (−1.31, −0.77)	−0.44 (−0.55, −0.33)
Model 3	Ref	−0.35 (−0.56, −0.14)	−0.70 (−0.93, −0.46)	−1.07 (−1.35, −0.79)	−0.45 (−0.58, −0.32)

Values are beta coefficients (β) and 95% CIs unless otherwise specified. Model 1: Adjusted for age, sex, and total energy intake. Model 2: model 1 + educational level, income level, smoking, drinking, physical exercise, and BMI. Model 3: model 2 + high blood pressure, high blood sugar, and prevalent CVD. IQR: interquartile range = 6.6.

**Table 4 nutrients-14-03548-t004:** Regression estimates and 95% CIs of eGFR per IQR increment in UPF intake according to participant characteristics.

Characteristic	Stratum	*β* (95% CI) ^a^	*p* for Interaction
Sex	Male	−0.51 (−0.83, −0.20)	<0.001
	Female	−0.03 (−0.04, −0.02)	
Age group	40–49	−0.41 (−0.61, −0.21)	0.080
	50–59	−0.43 (−0.65, −0.22)	
	60–69	−0.66 (−1.04, −0.28)	
Drinking	Non-drinker	0.10 (−1.95, 3.37)	0.129
	Past drinker	−0.07 (−0.80, 0.66)	
	Current drinker	−0.53 (−0.71, −0.36)	
Current smoker	No	−0.21 (−0.28, −0.14)	<0.001
	Yes	−0.81 (−1.23, −0.39)	
Regular exercise	No	−0.47 (−0.63, −0.3)	0.002
	Yes	−0.33 (−0.45, −0.21)	
BMI, kg/m^2^	<18.5	−0.03 (−0.35, 0.28)	0.002
	18.5–22.9	−0.12 (−0.21, −0.02)	
	23.0–24.9	−0.25 (−0.4, −0.09)	
	≥25.0	−0.39 (−0.56, −0.22)	
High blood glucose	No	−0.43 (−0.54, −0.32)	0.025
	Yes	−0.62 (−1.19, −0.04)	
High blood pressure	No	−0.34 (−0.44, −0.25)	0.681
	Yes	−0.39 (−0.6, −0.18)	
Prevalent CVD	No	−0.44 (−0.56, −0.32)	0.137
	Yes	0.44 (−0.55, 1.44)	

^a^ Per IQR increment in UPF intake. Adjusted for age, sex, total energy intake, educational level, income level, employment, smoking, drinking, physical exercise, BMI, high blood pressure, high blood sugar, and CVD. CVD: cardiovascular disease; BMI, body mass index.

**Table 5 nutrients-14-03548-t005:** Regression coefficients for the association between UPF sub-groups and eGFR.

UPF Sub-Groups (g/day)	*β* (95% CI) ^a^
Instant noodles	0.003 (−0.002, 0.009)
Breads	−0.007 (−0.012, −0.002)
Breakfast cereals and snacks	−0.028 (−0.042, −0.014)
Candies and chocolate	−0.168 (−0.234, −0.102)
Bread spreads (jam, honey, butter, and margarine)	−0.540 (−0.836, −0.238)
Meat and fish	0.004 (−0.028, 0.037)
Pizza and hamburgers	−0.021 (−0.035, −0.005)
Milk	−0.015 (−0.032, 0.002)
Yoghurt	−0.005 (−0.007, −0.003)
Ice cream	−0.031 (−0.039, −0.021)
Soymilk drink	−0.093 (−0.123, −0.063)
Soft beverages and fruit sodas	−0.003 (−0.005, −0.0001)
Sweet rice punch (“Sikhye”)	0.001 (−0.001, 0.003)

^a^ Adjusted for age, sex, total energy intake, educational level, income level, employment, smoking, drinking, physical exercise, BMI, high blood pressure, high blood sugar, and CVD.

## Data Availability

Data can be accessed upon request.

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
