# Peer review of "The Intake of Ultra-Processed Foods and Prevalence of Chronic Kidney Disease: The Health Examinees Study"

_nutrients, 2022, doi:10.3390/nu14173548_

Round 1

Reviewer 1 Report

Abstract should identify where the study was conducted, and when the data was collected. This should be included in the title if possible.

Proportion of UPF in the diet should be stated to be % of weight (if this is the case).

In line 20, the 10.6% increase should be identified as the IQR.

Identify what the PR is in the abstract.

The meaning of the ‘positive interactions’ need to be clarified, with the results of tests of significance for interaction given.

 Body of manuscript:

Line 49: this statement is a generalisation about UPFs which seems unlikely to be true in every case (for example, some UPF’s may have added vitamins). The reference for this sentence seems to suggest that its subject is ‘diets high in UPFs’ – please clarify.

Line 59: I don’t understand what the authors mean by ‘the additive interaction of UPF intake and selected demographic and lifestyle characteristics…’. The sentence might best express the aim of the study – i.e. “The aim of this study was to determine whether prevalence of CKD was associated with intake of UPFs in Korean adults, and whether this relationship differed by selected demographic and lifestyle factors [better to list these] using data from the 2004-2013 Health Examinees (HEXA) study.”

Line 65: an explanation of ‘invalid recruitment centres’ is needed.

Figure 1:  The text says that those aged 70 years and above (n=2626) were excluded, the figure refers to these as age > 69 and age < 40 – why not just age greater than 69 years as in the text?

‘Implausible dietary intake’ should be defined; missing covariates should be specified.

While the study sample is large, the basis for the collection is unclear – does it include people from all over Korea, or limited to one or few cities? The sample is described as population based (line 200) – what population, and did every member have an equal chance of selection?

Assessment of UPF intake: from the description, it is not clear which of the 106 food categories were defined as UPF and which were not. This could be clarified by providing a table of the FFQ categories in the supplementary materials. Additionally, an indication should be provided of how the 106 categories were combined to create 37 food groups for derivation of dietary patterns. The derivation of food patterns is surprising – this has not previously been mentioned (in the abstract, or in the study aims).

Line 110: while ‘activities that caused body sweating’ describes the intensity of an activity, it is not clear what the investigators considered ‘regular’ – was it once a week, once a month, etc. I note this is called ‘physically active’ in table 1, not ‘regular physical activity’.

Line 122/123: Aren’t these % (not frequencies) for categorical variables, and means (not least square means) for continuous variables?

Line 126: I don’t understand what is meant by quartile regression – could you clearly explain what is the dependent variable, and what are the independent variables? I don’t understand the results presented in table 4 at all.

Line 153-165: no descriptions of statistical tests, and no results of statistical tests – were these conducted?

Figure 2: The meaning of the figure is unclear. There is no labelling on the vertical axis, the horizontal axis is labelled ‘quantile level’ – perhaps quantile of UPF intake? The figure legend is not helpful.

Discussion:

The median %weight of UPF for this sample was 10.6% - the only other figure for Korea referenced was 25.1% (reference 12). This seems a large difference with no discussion – why are they so different?

Th apparent inverse relationship of dietary sodium with UPF % weight was notable in Table 1. This is mentioned in the discussion, but largely ignored in the next paragraph (lines 228-235) where discussion appears to be related to why sodium should be associated with both UPF intake and poorer renal function. It may be worth mentioning that FFQs are usually poor at measuring sodium intake in many populations (the validation study may give further detail.   

Further explanation of the additive interactions and their implications are needed – the interactions, while significant, look very small. Could they be entirely due to residual confounding? On the basis of these results, would the advice regarding UPF intake be different for older people, females, or fatter people? Should men, the elderly and the overweight/obese really be priority groups for intervention on the basis of these results, as suggested in the abstract?

The change in prevalence ratio changes little from model 1 to 3 as 15 covariates were added. The details we are shown of model 3 are little different to model 1 – did this surprise the authors? What was the purpose of including the final model with prudent dietary pattern and dietary acid load?

The use of the dietary patterns arising from the FFQ data is not clear – these variables are not discussed. What is the rationale of the authors for including these variables, and were any of the results using them informative?

Line 259: temporal relationships can only be inferred (rather than only temporal relationships can be inferred). It is true that the cross-sectional design is a significant weakness.

Author Response

Dear reviewer,

 We appreciate the time you devoted to review and provide constructive comments about our manuscript. We believe that the comments you provided have substantially improved the quality of this manuscript.

We have made efforts to address each comment point-by point and made several corrections as recommended. We look forward to receiving a positive verdict regarding this work.

Please see the responses to your comments below.

Authors

Reviewer 2 Report

This paper addresses an interesting topic on the relation between the intake of ultra-processed food and the prevalence of CKD. However, upon reading the manuscript, I have some concerns.

1.   Since the authors express eGFR as mean±SE. it means that eGFR showed normal distribution. Since the mean values of eGFR in all 4 subgroups were 93.2-96.7 mL/min, it seems that almost all subjects showed their eGFR values above > 60 ml/min. It is essential to show the proportion of CKD patients in all four groups in Table 1.  Furthermore, it would be better to show the association between the intake of ultra-processed food and eGFR expressed in continuous variables.

2.   Since eGFRcre is based on serum creatinine, eGFRcre is affected by muscle mass.  Since Q4 subjects were younger, it is possible that those subjects might have higher muscle components, leading to lower eGFRcre value is spite of similar BMI. Thus, it would be desirable to show the renal function by cystatin C-based eGFR or creatinine clearance.

3.   This study included DM, CVD and hypertensive patients.  These diseases impair renal function much stronger than intake of ultra-processed foods. Therefore, these patients should be excluded from the present study.

Author Response

(The authors gave the same response as above.)

Round 2

Reviewer 2 Report

This paper acquires quality  enough for publication in NUTRIENTS in its present form.